# Creep fronts and complexity in laboratory earthquake sequences illuminate delayed earthquake triggering

Sara Beth L. Cebry [1,7], Chun-Yu Ke [1,7], Srisharan Shreedharan[2,3,6], Chris Marone[2,4], David S. Kammer [5] & Gregory C. McLaskey [1]

Earthquakes occur in clusters or sequences that arise from complex triggering mechanisms, but direct measurement of the slow subsurface slip responsible for delayed triggering is rarely possible. We investigate the origins of complexity and its relationship to heterogeneity using an experimental fault with two dominant seismic asperities. The fault is composed of quartz powder, a material common to natural faults, sandwiched between 760 mm long polymer blocks that deform the way 10 meters of rock would behave. We observe periodic repeating earthquakes that transition into aperiodic and complex sequences of fast and slow events. Neighboring earthquakes communicate via migrating slow slip, which resembles creep fronts observed in numerical simulations and on tectonic faults. Utilizing both local stress measurements and numerical simulations, we observe that the speed and strength of creep fronts are highly sensitive to fault stress levels left behind by previous earthquakes, and may serve as on-fault stress meters.

Earthquakes are thought to rupture seismic asperities—sections of the fault that are stronger and more unstable than their surroundings either due to fault friction properties[1], geometry, or from locally high normal stress[2,3]. However, slower slip in the surrounding, weaker fault sections likely controls earthquake processes such as aftershock production, triggering, and days-to-decadal interaction between large earthquakes[4–6]. This slow slip can propagate in the form of fronts that separate relatively locked and creeping fault sections[7]. We broadly classify these slow fronts as "creep fronts". They exhibit slip speeds (<1 mm/s) and propagation velocities (<10 m/s) orders of magnitude slower than the dynamic rupture fronts that characterize regular earthquakes. Recent theoretical and numerical studies of such fronts have been applied to slow, post-seismic slip (afterslip) of large earthquakes[8,9] or their relation to underground fluid injection and induced seismicity[10–13]. The above models suggest that creep fronts typically slow down and attenuate as they propagate, with maximum slip velocity about four orders of magnitude slower than their

propagation velocity. They can transfer stress over distances that are larger than those expected by static coseismic stress changes[1]. While common in models, creep fronts have not been observed directly at seismogenic depths, but are seen in surface creep data[14], and inferred at depth from the migration of seismicity associated with fluid injection[15–17] and natural tectonic processes[5,18–21]. Slow slip events, often accompanied by tectonic tremor and low-frequency earthquakes[22–25], may be a spontaneously nucleating form of a similar slow process. Other studies of similar slow phenomena[26,27] have focused on explaining observations from laboratory experiments[28,29] of slow fronts that precede sliding of a frictional interface, similar to slow fronts long observed in association with the nucleation of dynamic rupture[30,31].

Heterogeneous fault properties that give rise to seismic asperities and interacting slow fronts are now widely recognized as crucially important for earthquake physics. Heterogeneity can influence fault strength and stability in complicated ways[32,33]. It is thought to increase

[1]School of Civil and Environmental Engineering, Cornell University, Ithaca, NY 14850, USA. [2]Department of Geosciences, Pennsylvania State University, University Park, PA 16802, USA. [3]University of Texas Institute for Geophysics, Austin, TX, USA. [4]Dipartimento di Scienze della Terra, La Sapienza Università di Roma, Roma, Italy. [5]Institute for Building Materials, ETH Zurich, Zurich, Switzerland. [6]Present address: Department of Geosciences, Utah State University, Logan, UT, USA. [7]These authors contributed equally: Sara Beth L. Cebry, Chun-Yu Ke. ✉e-mail: gcm8@cornell.edu

the likelihood of slow earthquakes[33–36], and it strongly influences earthquake initiation[30,37–39] and termination[40,41], and can therefore control the intensity, location, and magnitude of an earthquake, respectively.

Here, we report on multi-cycle interaction between slow fronts and seismic asperities leading to complex sequences of fast and slow laboratory earthquakes.

## Results

We have designed a large-scale laboratory experiment (Fig. 1a) where fault slip occurs within a shear zone composed of powdered quartz, known as a gouge. Quartz gouge friction is well characterized by the rate- and state-dependent friction (RSF) equations[42] that underpin a huge class of numerical models, increasingly used to help explain earthquake behavior ranging from specific earthquake sequences to whole catalogs of seismicity[43]. Recent RSF modeling studies[44] on a homogeneous fault of length W identified two nondimensional parameters that characterize fault behavior: 1: $R_u = W/h^*$, where $h^* = 2D_c G'/(\pi\sigma_N(b - a))$ is a critical elasto-frictional length[44], and 2: $R_b = (b - a)/b$, which describes the acuity of frictional weakening behavior. In the previous expressions, $G' = G/(1 - \nu)$, $\nu$ is the Poisson's ratio, $\sigma_N$ is normal stress, $G$ is the shear modulus, $D_c$ is the characteristic weakening distance and $b$ and $a$ are second-order RSF parameters[42]. With accumulated shear strain, granular gouge and fault rock evolve from velocity strengthening to velocity weakening friction and $D_c$ decreases[45]. As a result, $R_u$ increases as $h^*$ diminishes[46,47], shown by the gray arrow in

Fig. 1b (see also Supplementary Fig. 1 and Supplementary Table 1). These changes are due primarily to shear localization rather than reduction of particle size. Scuderi et al.[46] described this as an evolution from distributed deformation throughout the gouge layer to localized deformation along fault parallel shear planes, and showed evidence for comminution and grain size reduction. We expect similar behavior in our experiments since they utilize identical gouge layers (see "Methods"). The changes in friction increase $R_u$ and cause the sample to transition from steady creep to progressively more complex behavior (Fig. 1b), consistent with RSF models[44]. The categories of fault behavior in $R_u$–$R_b$ parameter space in Fig. 1b are based on the homogeneous model of Barbot[44]. Similar results were obtained by Cattania[48]. Our sample with heterogeneous properties, described below, produces a qualitatively similar behavioral progression to the homogeneous numerical simulations[44,48], but at lower $R_u$ levels.

RSF parameters are determined from laboratory experiments on small samples treated as a single-degree-of-freedom (SDOF) system[42,46,47], but our sample, like tectonic plates, behaves as a deformable continuum. In our experiments, the fault gouge is held between two 760 mm-long blocks of poly(methyl methacrylate) (PMMA), a glassy polymer about 15 times more compliant than rock ($G_{PMMA} \approx 1$ GPa, $G_{rock} \approx 15$ GPa). PMMA and other compliant materials are frequently utilized for earthquake rupture experiments[28,29,49,50], in part because of their small $h^*$ compared to rock. They have only recently been combined with geologically realistic fault gouges[37,38,51] that compact, strengthen, and evolve with continued shear slip. Our

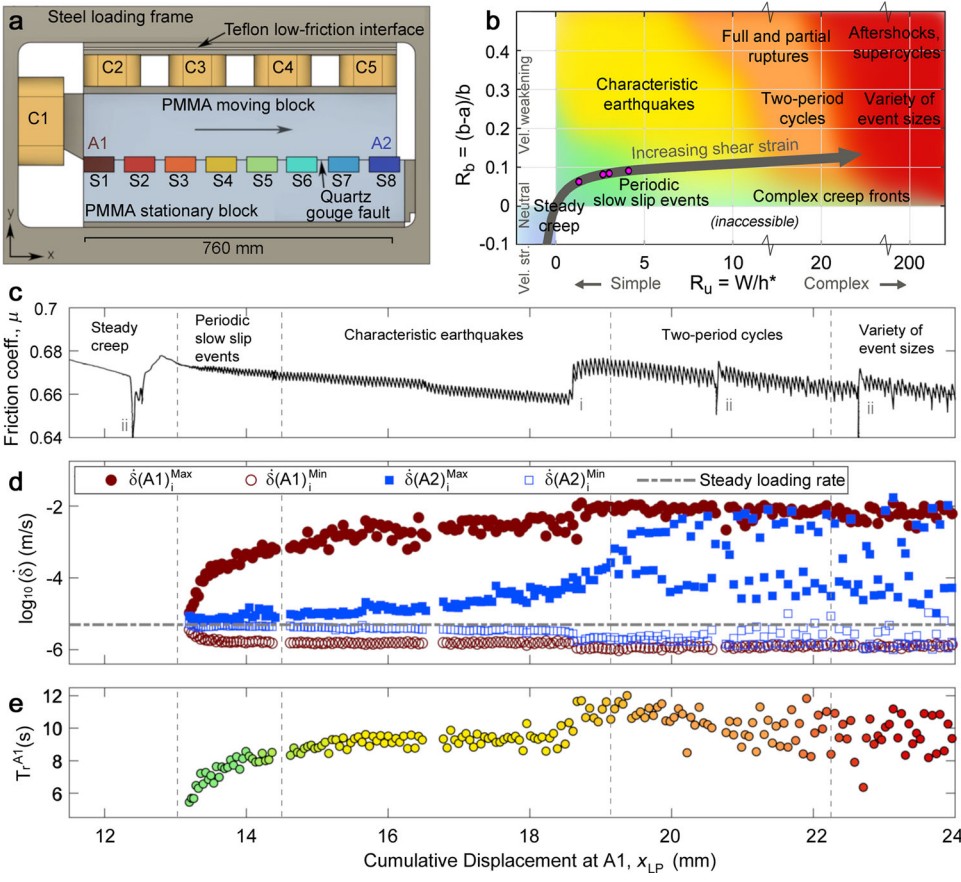

**Fig. 1 | Experimental system and evolution of earthquake behavior. a** Diagram of the sample loaded with hydraulic cylinders C1-C5. 8 slip sensors (S1-S8) are color-coded: forcing end (A1, red) to leading end (A2, blue). **b** In our experiment, friction properties evolve with continued shear strain (Supplementary Table 1 and Supplementary Fig. 1) and chart a path (gray arrow) through $R_u$–$R_b$ space that transitions from steady creep to more complicated behavior. Magenta circles denote locations of four examples shown in Fig. 2. **c** Sample-average friction coefficient μ. Annotations mark unload-reload cycles (i) and holds (ii) where the sample rested in essentially stationary contact. **d** Maximum and minimum slip rate at either end of the sample (A1 and A2) plotted every A1 stick-slip cycle. **e** A1 recurrence time $T_r^{A1}$.

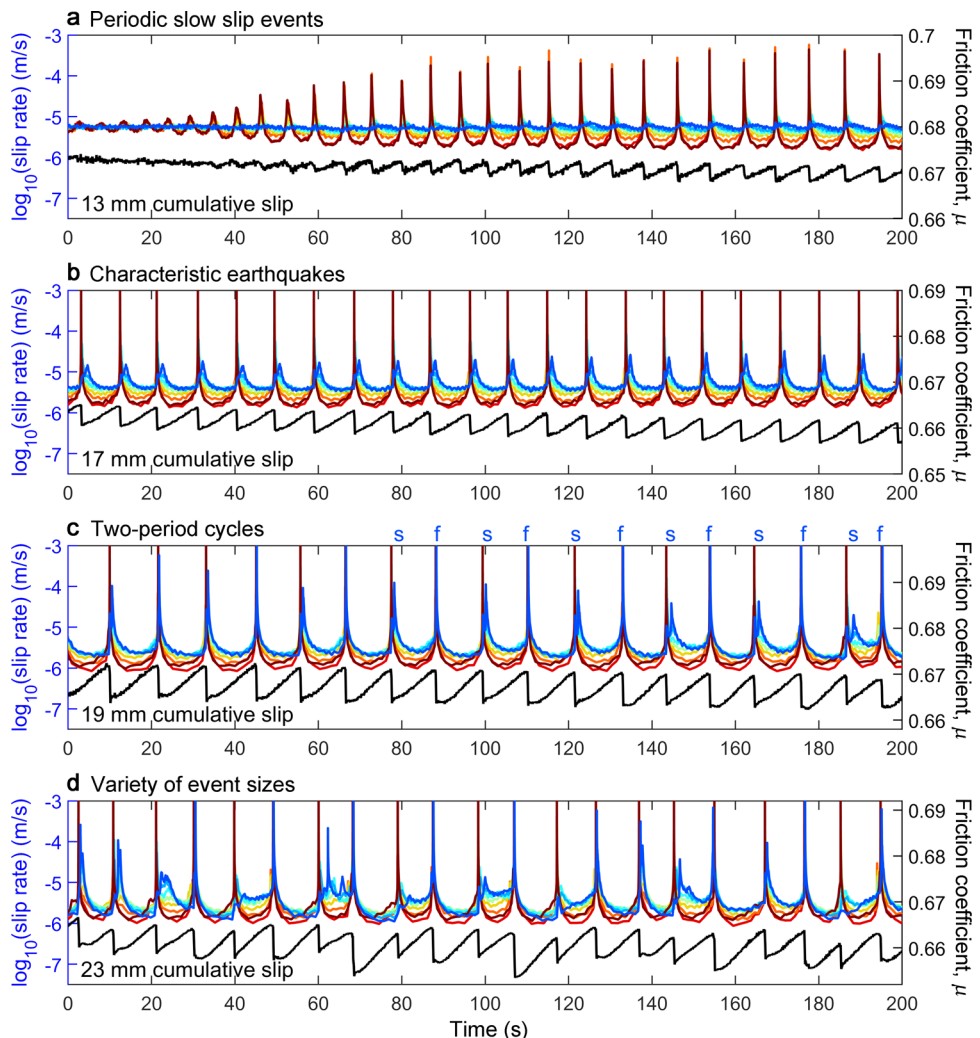

**Fig. 2 | Examples of sequences from four displacement intervals showing a progression from simple to complex.** Data from the experiment of Fig. 1. Sample-average friction (black) and slip rates (colors) at 8 locations from A1 (red) to A2 (blue) (see Fig. 1a for locations). **a** Slow slip events grow on A1 while A2 slips steadily ($x_{LP}$ = 13 mm). **b** Identical, periodic seismic ruptures on A1 with slowly evolving slow slip events on A2 ($x_{LP}$ = 17 mm). **c** A bifurcation wherein A2 oscillates between progressively faster (annotated f) and slower (annotated s) events ($x_{LP}$ = 19 mm). **d** Complex sequences with variable recurrence interval, stress drop, and slip rates on A1 and A2 ($x_{LP}$ = 23 mm).

work demonstrates that this combination can be used to make a realistic scale model of earthquake interactions, including post-seismic slip and triggering. The sample behaves like 10 m of rock (760 mm * $G_{rock}/G_{PMMA} \approx$ 10 m). Similar sequences on 3 m rock samples[30,40], show primarily characteristic events with far less complexity. SDOF laboratory experiments achieve complex behavior only by tuning stress and friction parameters to match the resonance of the loading machine and do not include spatial variations in behavior over the sample dimensions[46,47,52]. Here, however, such spatiotemporal complexity is a natural result of the fault length and its frictional properties and heterogeneity.

## Evolution from simple to complex

In this work, the gouge layer is prepared with uniform initial thickness and composition and loaded with 10 MPa average normal stress $\sigma_N$ (Methods). We shear the sample at a constant rate of 6 μm/s, to roughly simulate tectonic loading, and we measure the increasingly complex behavior that naturally develops as a function of cumulative fault slip $x_{LP}$. The experiment exhibits two main seismic asperities at the forcing end (A1, x = 0) and leading end (A2, x = 0.76 m) of the sample (Fig. 1a). We describe the sample ends as seismic asperities because they are more unstable than their surroundings. The

asperities have locally high $\sigma_N$ due to a mechanical edge effect common to biaxial sample configurations[53–55] (Supplementary Fig. 2). This causes them to accumulate higher shear stress (Supplementary Figs. 17 and 18), and slip faster with greater stress changes than the center part of the sample. The free surface at the end of the sample also reduces their stiffness and enhances instability. Note that the above definition is different from micron-scale junctions that compose a frictional interface (e.g., ref. 56) that are sometimes also referred to as asperities.

The entire sample slips stably at first, then, at $x_{LP}$ = 13 mm A1 begins to produce slow slip events that grow larger and faster while A2 continues to slip stably (Figs. 1c, d and 2a). From then on, A1 slip events occur quasi-periodically with recurrence interval $T_r^{A1}$. We catalog all events by measuring the maximum and minimum slip velocities at both A1 and A2 every $T_r^{A1}$ (Fig. 1 and Supplementary Fig. 3). From $x_{LP}$ = 13 to 19 mm, A1 events grow progressively faster with increasing shear displacement and begin to radiate detectable seismic waves while their stress drops steadily increase. Stress drops from A1 events drive creep fronts that gradually become more defined (Supplementary Fig. 4). A2 behavior transitions into a set of slow slip events that become progressively faster (Figs. 1d and 2b), and the time delay between A1 and A2 events steadily decreases (Supplementary Fig. 5).

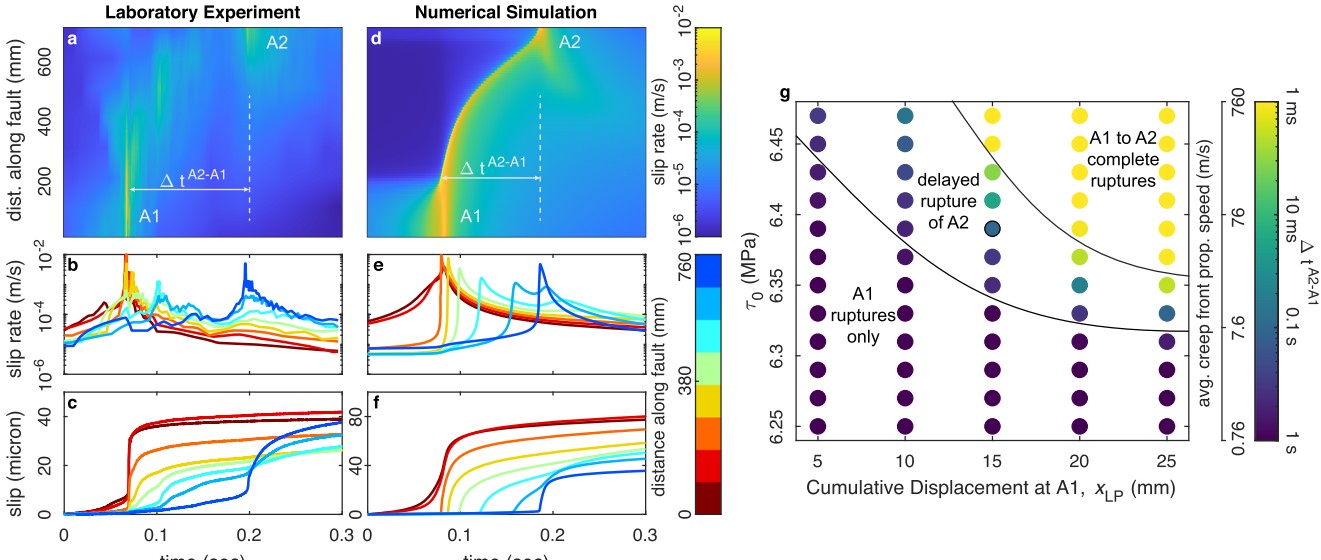

**Fig. 3 | Creep front dynamics in the experiment and numerical simulations.**
**a**–**c** Example of A1-to-A2 delayed triggering observed at $x_{LP}$ = 23 mm using local slip measurements. **d**–**f** RSF numerical simulation shows a well-defined creep front that both slows down and attenuates as it propagates. **g** Each dot is a separate RSF

numerical simulation to explore creep front triggering time from A1-to-A2, $\Delta t^{A2\text{-}A1}$ (note color scale) as a function of friction parameters that depend on a $x_{LP}$ (see Supplementary Table 1) and $\tau_0$, the initial stress for the region between A1 and A2 (see Supplementary Fig. 11).

At $x_{LP}$ = 19 mm, A2 undergoes a bifurcation. A2 oscillates between fast slip events (>10 mm/s) driven by fast creep fronts and slow events (100 μm/s) driven by more sluggishly propagating creep fronts (Fig. 2c). After about 20 mm of cumulative fault slip, A2 events transition to more aperiodic and chaotic behavior (Fig. 2d). Creep fronts are occasionally so slow and weak that a second A1 event occurs before A2 ruptures. In some cases, we observe creep fronts that propagate in the opposite direction, from A2 to A1, to influence and in some cases directly trigger subsequent A1 ruptures (Supplementary Fig. 6). These asperity feedback mechanisms increase the variation in both $T_r^{A1}$ (Fig. 1e) and sample average friction coefficient μ (Fig. 2d).

## Creep fronts in experiments and numerical models

We describe the aseismic slip transients as creep fronts; however, they are markedly different from the smooth fronts depicted by numerical simulations (Fig. 3), described below. In the experiment, the creep front includes transient increases in local slip rate due to smaller events that rupture secondary asperities (Fig. 3a and Supplementary Fig. 6). The gouge layer and plastic blocks were prepared as uniformly as possible, but secondary asperities likely developed from small heterogeneity in the initial gouge distribution. Their spatial locations persist over many stick-slip cycles (Supplementary Figs. 6 and 7), similar to observations on natural faults[23]. Feedback between small seismic events and the slow slip that links them[2] may produce creep fronts that are more variable and complex than smooth numerical simulations would suggest[1,11,12]. Furthermore, the peak slip velocities and seismic radiation associated with the rupture of A1, A2, and secondary asperities grew stronger throughout the experiment, (Fig. 1d and Supplementary Fig. 8), so we suggest that asperities develop naturally as part of the evolving fault fabric and stress redistribution. Previous work shows that higher $\sigma_N$ causes a more rapid transition to unstable friction behavior (smaller $D_c$, larger b-a) with continued shear[46]; thus, the structures that produced A1 and A2 may also drive changes to friction properties that reinforce them[57].

To further study the creep fronts, we developed a set of numerical models that probe delayed triggering at the laboratory scale. The models utilize a 2D spectral boundary integral method with spontaneous initiation and fully dynamic rupture propagation ("Methods"). Slip along the interface is governed by RSF. Our models are highly

simplified; we define the two asperities (A1 and A2) as regions with locally high shear stress, while normal stress and friction properties are constant across the domain. As a result, the models are not used to reproduce multiple cycles; each model's initial conditions produce only one rupture of A1 and A2. The specific size and stress state of two modeled asperities were tuned to yield spontaneous rupture of A1 followed by delayed rupture of A2 (Fig. 3 and Supplementary Figs. 9 and 10), but specific asperity characteristics likely differ from those in the experiments (see "Methods", Supplementary Figs. 11 and 12).

Despite their simplifications, the models provide insights into how creep front speed is affected by friction properties and initial stress levels. Keeping stress levels and geometry of A1 and A2 constant throughout all models, we studied the triggering behavior as a function of the shear stress level ($\tau_0$) between the two asperities and changes in friction parameters with shear of the evolving gouge layer (Fig. 3g), which are well constrained by previous studies on gouge layers of identical thickness and composition as those we study[46,47] (Supplementary Fig. 1 and Supplementary Table 1). The simulations show that small changes in $\tau_0$ (50 kPa or <1%) cause two orders of magnitude variation in the average propagation speed of the creep fronts. Fronts propagate faster at higher $\tau_0$, consistent with other recent studies[10–12,58,59]. Furthermore, as the fault fabric develops, and a-b and $D_c$ decrease and $R_u$ increases[46], creep front behavior and the isolated events at A1 and A2 become increasingly sensitive to small variations in stress levels (Fig. 3g). This is the likely reason for the increasingly complex behavior observed late in the experiment, despite highly periodic behavior early on (Fig. 1).

## Local stress measurements

To directly test if laboratory creep front characteristics and triggering times also show stress dependency, we repeated the experiment with strain gages collocated with the slip sensors (S1–S8 in Fig. 1a) so local shear stress could be monitored. This second experiment reproduced all of the main characteristics reported above, with minor differences (Supplementary Fig. 13). We observed that the passage of the creep front is marked by a drop in local shear stress and a peak in slip rate (Fig. 4a), consistent with numerical simulations[8]. We use the latter feature to estimate the creep front velocity $v_{cf} = d/(\Delta t^{S7\text{-}S6})$, where $\Delta t^{S7\text{-}S6}$ is the creep front travel time between sensors S6 and S7 (Fig. 4a) and

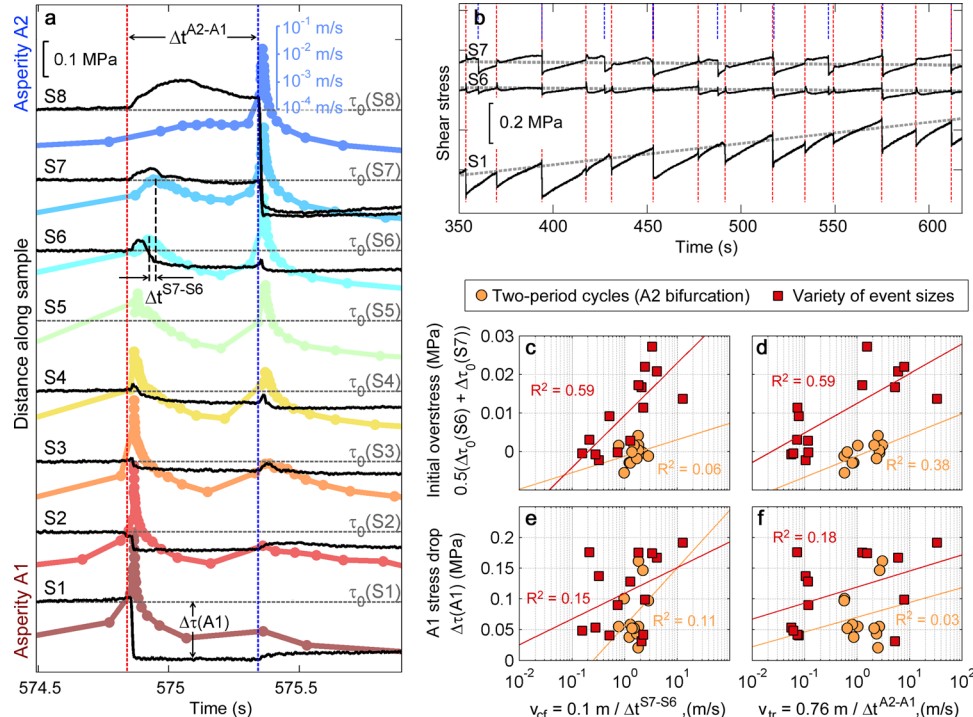

**Fig. 4 | Creep front sensitivity to local stress. a** Local shear stress changes (black lines) derived from strain gages S1–S8 (Fig. 1a) alongside local slip rate (colors) plotted on a log scale and offset by sensor location along the fault. Gray horizontal dashed lines mark $\tau_0$ prior to the A1 event (for black lines) and also mark a $10^{-4}$ m/s slip rate reference line (for colors). Slip events begin at A1 (note stress drop) and attenuate as they propagate towards A2. Note the reduction in slip rate from S1-S8 and transition from stress drop (red vertical line) to stress increase that eventually triggers slip at A2 (blue vertical line). **b** Stress changes over many cycles of A1 events (red vertical lines) and A2 events (blue vertical lines) show progressive strengthening (at gage S1) and weakening (at gages S6, S7). **c–f** Front velocities $v_{cf}$ and $v_{tr}$ derived from parameters in (**a**) (see text) are correlated with initial overstress $\Delta\tau_0$ (stress relative to gray trend lines in **b**), but exhibit little correlation with stress drop from A1 events that initiate them. Orange circles and red squares are data from different stages of the experiment as shown in Supplementary Fig. 13, and $R^2$ are labeled.

$d = 0.1$ m is the distance between sensors. We also use the time delay between A1 and A2 events ($\Delta t^{A2\text{-}A1}$) to calculate the average triggering velocity $v_{tr} = W/(\Delta t^{A2\text{-}A1})$, which depends on both the creep front propagation and A2 nucleation time, and is more comparable to creep front speed inferred from migrating seismicity[18,19]. These front propagation velocities $v_{cf}$ and $v_{tr}$ range from about 0.1–1 m/s and are consistent with the range of slow slip propagation speeds inferred in the Cascadia subduction zone[24,25].

For the lab experiments, we compare creep front characteristics to the initial overstress $\Delta\tau_0$, not the absolute stress levels $\tau_0$, since different fault locations were observed to progressively strengthen or weaken with cumulative slip (Fig. 4b). Overstress is the shear stress level relative to the fault's strength when sliding at a constant rate (steady state). Over the course of an earthquake cycle or stick-slip cycle, the fault transitions from being above steady state (prior to the earthquake) to below steady state (just after an earthquake), and this transition is facilitated by healing and breaking of frictional contacts[60]. We define $\Delta\tau_0 \propto \tau_0 - \tau_{ss}$, where $\tau_{ss}$ is a reference level that changes linearly over time to match the long-term local strengthening or weakening trend over >10 stick-slip cycles (gray trend lines shown in Fig. 4b). Note that in the numerical models described previously $\tau_0 \propto \Delta\tau_0$, since initial slip velocity $V_{ini}$ and other parameters are uniform across the model.

Comparing many events from an experimental sequence, we find that $v_{cf}$ and $v_{tr}$ are both correlated with the estimated overstress $\Delta\tau_0$ prior to A1 events (Fig. 4c, d), and that 20 kPa variation in $\Delta\tau_0$ can cause an order of magnitude velocity change, consistent with our numerical models. In contrast, we observe little correlation between $v_{cf}$ and $v_{tr}$ and the stress drop of the A1 events that initiated the creep front (Fig. 4e, f).

The sensitivity to $\Delta\tau_0$ explains the oscillatory behavior of the A2 bifurcation observed at $x_{LP} = 19$ mm: stronger A2 events reduced stress levels more significantly so the subsequent creep front propagated slowly and triggered weaker A2 events. Weaker A2 events have smaller stress drop and thus do not reduce stress levels as much, which primes the fault for faster subsequent creep fronts and more rapid triggering of stronger events. This relationship was also deduced from the strength and timing of A1 and A2 events (Supplementary Fig. 14), but can be easily obscured by complex interactions between A2 and A1, such as back-propagating creep fronts (Supplementary Fig. 6) that affect the timing of A1 events. In the above discussion, we refer to "strong" events as those that slip faster, have larger total slip amount, and have larger local stress changes. These parameters are directly correlated for events with rupture dimensions less than ≈5 h* ref. [61].

Garagash[12] showed that when creep front propagation distance $L/L_b$ is small, the front velocity is affected by the hypocentral forcing, which, in our case, is the stress drop of the A1 event that initiated it. When the front propagates farther ($L/L_b > \approx 5$), its speed and strength become dominated by the initial fault overstress $\Delta\tau_0$, (Supplementary Fig. 15). In previous expressions, $L_b = D_c G'/(\sigma_N b)$. In our experiments, $L_b$ decreases with continued shear so $L/L_b$ increases. Earlier in our experiments ($x_{LP} < 19$ mm), the rupture of A1 and A2 are synchronized[1] and driven by A1. The sample behavior is highly regular. The A2 bifurcation occurs at $x_{LP} = 19$ mm (Fig. 1d), likely due to a crossover from hypocentral forcing-dominated creep fronts ($L/L_b < \approx 5$) to $\Delta\tau_0$-dominated creep fronts ($L/L_b > \approx 5$), (see Supplementary Table 1). Thus, later in the experiment ($x_{LP} > 20$ mm), the friction parameters have evolved such that the triggering time of A2 depends more on $\Delta\tau_0$ left by previous A2 ruptures than by the A1 events. The result is highly variable sample behavior.

The strain and slip measurements also help confirm expectations from models (Supplementary Fig. 2), indicating that the fault sections with high $\sigma_N$ that create asperities A1 and A2 are likely just a few cm in size, near the sample ends. For example, the gradual drop in shear stress and local maximum in slip velocity measured at S8 (at 575.20 s in Fig. 4a) coincident with the passage of the A1-to-A2 creep front clearly precede the rapid drop in stress (575.35 s) associated with dynamic rupture of A2. This suggests that the creep front passed by this measurement location−3 cm from the leading edge of the sample−on its way to the A2 asperity. Thus, A2 is likely smaller than 3 cm. Despite such localized asperities, dynamic rupture and rapid postseismic slip extends 20−30 cm (e.g., S1−S3 at 574.8 s in Fig. 4a), affecting the stress levels in this larger region, similar to RSF modeling results[3].

## Discussion

Asperities are responsible for complex slip distributions often observed within dynamic earthquake rupture; however, the asperity interactions we observe involve both dynamic slip and slow fronts and are likely relevant to faults that exhibit a mixture of seismic and aseismic slip such as some subduction zones and mature plate boundary faults[5,19–21,36]. Our work utilizes a hybrid sample, whose elasticity is controlled by compliant plastic forcing blocks but whose friction is dictated by a shear zone composed of geological material (quartz gouge). This creates complicated earthquake interactions at length scales $L > 5L_b$ that would normally only occur on rock samples 10 m in length. The laboratory experiments demonstrate creep fronts initiated by slip instabilities; the fronts' hypocentral forcing is an earthquake rather than fluid injection[10–12] or a spontaneous slow process such as earthquake nucleation or a slow slip event[30,31,62]. Many properties of the fronts are consistent with numerical simulations and theory[8], including the linear relationship between maximum slip velocity and propagation velocity (Supplementary Fig. 16). However, the experimentally observed creep fronts are not as smooth or as sharply defined and often include variations in slip rate due to heterogeneous fault properties (Fig. 3a–c). Also different from many simulations, our creep fronts propagate in a mildly velocity-weakening friction, similar to some inferences from subduction zones[6]. Our observed front propagation speeds range from 0.1 to 10 m/s, broadly consistent with slow slip speeds inferred from migrating tectonic tremor sources[24,25]. Some creep fronts detected on deep sections of the San Andreas fault are triggered by seismic waves and propagate somewhat faster (10−30 m/s)[21]. The higher propagation speeds and the readiness for failure implied by such "triggerability" are both consistent with a high fault overstress $\Delta\tau_0$, described below.

We find that outside a distance $\approx 5L_b$ from the location of hypocentral forcing, creep front strength (maximum slip velocity) and propagation velocity are extremely sensitive to fault stress levels in excess of the steady sliding strength (fault overstress $\Delta\tau_0$). Our experiments also demonstrate how small asperities can affect the stress state in larger fault regions surrounding them (Fig. 4a). Thus, creep front propagation speeds in the surrounding regions can be an indicator of the conditions of the nearby asperities. Put simply, front propagation speed is sensitive to a fault's readiness to host an earthquake rupture.

Extending these ideas[12], we suggest that faults that are more strongly velocity weakening and/or those with significant overstress may host fast creep fronts that can quickly accelerate into subsequent dynamic rupture and may only be identified seismically as a 1–10 s pause between a triggering event and its (potentially larger) aftershock[63,64]. Faults that are strongly velocity strengthening will produce creep fronts with more limited spatial extent[8], where asperities act in isolation and are less likely to trigger neighboring earthquakes on days-to-years timescales[4]. Faults that are nearly velocity neutral with modest overstress might host extended creep fronts that trigger seismicity in a migrating pattern that can be observed over days

to weeks[5,18–20]. For example, many slow slip events are detected in shallow subduction settings where sampled material has been shown to be nearly velocity neutral and contain phyllosilicates whose weakness would limit overstress[36].

## Methods

### Experiments

We study shear within a layer of quartz gouge separating a PMMA moving block (762 mm × 203 mm × 76 mm) and a PMMA stationary block (787 × 152 × 76 mm) (Fig. 1a). The simulated fault is 762 mm × 76 mm with area A = 0.0579 m². The gouge layer, composed of dry MIN-U-SIL-40 99.5% $SiO_2$ grain size 2–50 µm mean size 10−15 µm, was prepared 5 mm thick on the stationary block, placed at 95% relative humidity for 24 h, then sandwiched between the PMMA blocks, loaded into the apparatus, and compacted to 2.5 mm thickness. Small teeth, 1 mm deep with 1.1 mm spacing, were machined into the fault faces of each PMMA block to ensure that the principal slip surface was within the gouge layer rather than the plastic-gouge interface (Supplementary Fig. 18). Care was taken to ensure that the teeth and sample preparation procedure were identical to previous experiments used to measure friction properties and gouge microstructure[46,47]. The sample was loaded in a direct shear biaxial apparatus shown schematically in Fig. 1a and used in previous studies[65,66]. Hydraulic cylinders C2-C5 (Fig. 1a) apply $\sigma_N = 10$ MPa, held essentially constant for the entire experiment by closing a valve. Cylinder C1 shears the sample at a constant rate of 6 µm s⁻¹. Slip between the PMMA blocks was measured with 8 eddy current displacement sensors located along the length of the fault at locations 30, 130, 230, 330, 430, 530, 630, and 730 mm from the forcing end (Fig. 1a). Slip also occurs on a Teflon low-friction interface ($\mu_{Teflon} \sim 0.1$) located between C2−C5 and the steel loading frame, and this redistributes shear stress from the forcing end to the entire sample. Reported values of µ correspond to $\mu_{gouge}$ where $\mu_{meas} = \mu_{gouge} + \mu_{Teflon}$ was determined from the sample-average shear stress $\tau$ and $\sigma_N$, determined from hydraulic pressure. An array of four piezoelectric sensors (Panametrics V103) glued to the PMMA, 38 mm from the fault, detect vibrations (Supplementary Fig. 8). We conducted a suite of experiments varying $\sigma_N$, gouge layer mineralogy, and loading procedure (Supplementary Table 2) and report here on one representative experiment on pure quartz gouge (QS04_021, Figs. 1−3 and Supplementary Figs. 4−8 and 14) and a nearly identical follow-up experiment (QS04_023, Fig. 4 and Supplementary Fig. 13) that included strain gage pairs located 5 mm from the gouge-filled fault. We distribute a brand-new gouge layer for every experiment. All six experiments on quartz showed a similar evolution of behavior that culminated with partial ruptures and variable delayed triggering, though not all experiments were loaded at a smooth and constant rate or instrumented as completely. Talc produced only slow slip, and the behavior of gypsum was less repeatable.

### Numerical model

Assuming the fault behaves uniformly across the thickness (z direction), the fault is represented as a mode II crack in 2D spectral boundary integral method simulations[67,68] with spontaneous initiation and fully dynamic propagation. The fault and slip are both in the x direction. Supplementary Fig. 11 shows the parameterized initial stress distribution $\tau_{ini}(x)$. The two asperities are represented by local patches with high $\tau$. In the experiments, normal stress is higher at the sample ends (Supplementary Figs. 2, 17, and 18), and this is the reason for the high shear stress there; however, in the model, $\sigma_N = 10$ MPa is uniform across the entire domain to provide uniform RSF and simplify the model considerably. This simplification is only permissible because each simulation only consists of one set of A1 and A2 ruptures. The simulation domain is twice the size of the domain of interest ($0 \le x \le L$, where $L = W = 0.76$ m), and A1 and A2 are regions with half-length $r_1 = 0.2$ m and $r_2 = 0.05$ m and locally high shear stress $\tau_1 = 6.56$ MPa

and $\tau_2 = 6.48$ MPa, respectively. Initial shear stress outside the domain of interest $\tau_{ext} = 5.5$ MPa for all models, whereas the initial stress between two asperities $\tau_0$ varied for different simulations from 6.25 MPa to 6.47 MPa (Fig. 3g). Slip along the interface is governed by rate- and state-dependent friction (RSF) equations with slip law formulation[69–71]. RSF parameters: a, b, and $D_c$ were determined with SDOF experiments on identical gouge layers[46,47] with similar sample preparation between the smaller and larger experiments (reported in Supplementary Table 1 and Supplementary Fig. 1). Elastic material properties, domain size $W$, loading rate $\dot{\tau} = 0.08$ MPa/s, and initial slip velocity along the fault, $V_{ini}$, match the experiment. Those properties and $\tau_1$ and $\tau_2$ are the same for all simulations. For each simulation, friction parameters (a, b, $D_c$), $\sigma_N$, and $V_{ini}$ are applied uniformly across the whole domain, and the state variable $\theta$ is then initialized by $\theta(\sigma_N, \tau_{ini}, V_{ini})$ at each location to enforce the equilibrium of the RSF equation. During the simulation, the shear stress is uniformly increasing, i.e., $\tau(x, t) = \tau_{ini}(x) + \dot{\tau}t$. The finite dimensions of the experiment in the y direction likely have only minor effects on the stress transfer at fault, with ~5% error compared to the infinite elastic space assumed in the model; however, the model neglects the free surface boundary conditions at the sample ends, which tend to hasten creep front propagation and increase the strength of asperity rupture (Supplementary Fig. 12). As a result, the asperity sizes in the model ($r_1, r_2$) are larger than expected in the experiment.

## Data availability

The experimental data generated in this study are freely available on eCommons via https://doi.org/10.7298/4rmf-w308. Simulation data from this study are freely available at https://doi.org/10.3929/ethz-b-000568751.

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

## Acknowledgements

The authors thank A. Zhang, K. Sorhaindo, and T. Brock for assistance with sample preparation. This work was supported by US National Science Foundation grants EAR-1763499 (G.C.M., C.-Y.K., and D.S.K.) and EAR-1847139 (S.B.L.C. and G.C.M.), and EAR-1763305 (C.M.), European Research Council Advance Grant 835012 (TECTONIC) (C.M.), US Department of Energy grants DE- SC0020512 and DE-EE0008763 (C.M.).

## Author contributions

D.S.K., G.C.M., and C.M. devised the study; S.B.L.C., C.-Y.K., and G.C.M. performed the experiments; S.S and C.M. assisted with gouge preparation and friction parameters; S.B.L.C and G.C.M. analyzed the data; C.-Y.K. performed the numerical modeling; C.-Y.K., D.S.K., and G.C.M. analyzed the simulation data; G.C.M. wrote the manuscript with contributions from all authors.

## Competing interests

The authors declare no competing interests.
