## [Peer Review File · Nature Communications]

Editorial Note: This manuscript has been previously reviewed at another journal that is not operating a transparent peer review scheme. This document only contains reviewer comments and rebuttal letters for versions considered at Nature Communications. [Mentions of prior referee reports have been redacted.]

REVIEWER COMMENTS

Reviewer #5 (Remarks to the Author):

Review of "Creep fronts and complexity in laboratory earthquake sequences illuminate delayed earthquake triggering" by Cebry et al. submitted to Nature Communications

This paper describes laboratory experiments that simulate a transition of fault slip behaviors with shear displacement and a delayed triggering on a gouge-filled laboratory fault. The experimental data and a numerical simulation suggest that velocity of creep front causing the delayed triggering is sensitive to shear stress level.

I enjoyed reading the manuscript very much. I think that this study is designed well and original. The experimental setting is noble, and the obtained data is valuable. However, I have some concerns as below, which should be addressed before publication.

[Major comments]

1. Asperity

1-1. Definition of asperity

As well known, definition of the "asperity" differs depending on the research theme (e.g. the area where really contacts, the area radiating the high-frequency seismic wave, the area slipped a lot during the earthquake, etc.). Therefore, unless it is clearly defined first, the readers including me cannot decide whether the authors' data interpretation and the model based on it is reasonable or not. It seems that the authors treat the high "normal" stress region as the asperity based on the FEM calculation (Supplementary Fig. 2), while the high "shear" stress is assumed for the asperity in the numerical simulation without any reasoning. Although the assumption should be basically reasonable as far as the friction coefficient is uniform over the fault, confusion and/or misunderstanding can be caused unless detailed explanation is added. To avoid such troubles, the authors should clearly define the physical meaning of the asperity, including the second asperity in L131 and others, in this study before starting the discussion.

1-2. Evidence for definition

Related to 1-1, direct evidence for the definition of the asperity should be provided. Although the authors may argue that the local normal stress cannot be measured, I believe that they can show the local shear stress distribution by using the strain gauge data, which is necessary for the readers to accept the authors' model. I also think that the strain data can help us understand what the second asperity is.

2. Dependence of creep propagation speed on shear stress level

I understand that the main message from this study is that the creep propagation speed is sensitive to the shear stress level, which is individually suggested from the numerical simulation and the experimental data. But I have concerns about each of them.

2-1. Numerical simulation with simplified model setting

I have questions about the necessity of the numerical simulation in this study and the validity of the conclusions based on it. I agree that it is generally difficult and not always necessary to construct a perfect model which takes account of all the conditions to reproduce the real fault slip behaviors in the numerical simulation. However, it must be noted that we don't know how universal the calculated results are under the limited conditions. Regarding the current numerical simulation, the authors just investigate the creep propagation speed while varying only the shear stress level (at a X_{LP}). So, it's not surprising that those correlation was seen. For the universal conclusion, dependence of the creep propagation speed on other parameters should also be investigated. For example, although the asperities at both edges of the fault are modeled as the regions with locally high shear stress, the normal stress should also be high there as shown in Supplementary Fig. 2 and actually the high normal stress should cause the local high shear stress. According to the definition of L_b , higher normal stress makes L_b smaller, which should lead to more unstable. Therefore, I wonder that the normal stress rather than the shear stress intrinsically characterizes the asperity and also controls the characteristics of the slip there (e.g. slip amount, slip rate, etc.). Since it should be natural to consider that the slip at the asperity affects the following creep behaviors, the normal stress may finally control the creep propagation speed.

2-2. Overstress τ_0 in experiment

I don't understand why the deviation of the local shear stress from its long-term trend is defined as the overstress τ_0 in the experiment data. What is the physical meaning? Why isn't the absolute value used like the numerical simulation? I also wonder how the authors determined the gray trend lines shown in Figure 4b. Instead of the current one, how about using the deviation of the local shear stress at S6 and S7 from the temporal mean of the shear stress over the fault? I also suggest the authors to plot the shear stress data at S6 in Figure 4b because it is used as the parameter in Figure 4c.

3. Transition of slip behavior

One of the key features of this experimental study is observation of the transition of the slip behaviors with the shear displacement. The authors attribute it to the change in the frictional parameters with the evolution of shear fabric within the gouge layer. Although I can basically agree with the interpretation, I also have a concern. I think that a slip initiating at the asperity A1 must reach the forcing end of the fault. Once the slip reaches the end of the fault, the following slip behavior interacts with the apparatus system. Since the shear load is applied with the hydraulic jack, the oil volume within the jack increases with the shear displacement, which can greatly reduce the stiffness of the hydraulic jack and then leads to the decrease in the whole stiffness of apparatus. I just wonder that such decrease in the apparatus stiffness with the shear displacement can cause an apparent transition of the slip behavior.

4. Gouge layer

4-1. Uniformity of gouge distribution (L274-277 "The gouge layer ... 2.5 mm thickness.")

Given the significance of the effect of the gouge distribution on the slip behavior, it is necessary to describe how uniformly the gouge was distributed on the fault. Even with a small excess of the gouge, local normal and shear stress in the region can become higher than those in the

surrounding region, which will form an asperity. I just suspect that the second asperity was generated by such a small heterogeneity of the gouge distribution.

4-2. Microstructure of gouge layer (L277-280 "Small teeth, ... plastic-gouge interface.")
Because the authors consider that the slip behaviors in the current experiment are governed by the frictional properties of the gouge, it is important to confirm that the slip did not take place at the interface between the PMMA block and the gouge but within the gouge layer. Did the authors investigate the microstructure of the sheared gouge layer after the experiment? Please show it if yes.

[Minor comments]

5. L106-107 "The entire sample ... (Fig. 1c, 1d; Fig. 2a)."

Please clarify the meaning of "bifurcation" here. The "bifurcation" for the asperity A1 in L106 and that for A2 in L117 are the same thing? I think that the second one means the alternating slip styles between slow and fast on an asperity, which should be consistent with the slip styles called "bifurcation" in other studies (e.g. Veedu et al., 2020, GRL). But I don't understand what the authors exactly mean by that term in L106.

6. L141-142 "the structures that ... that reinforce them."

What are "rheological changes"? Does it mean the changes in the frictional parameters? Please add some explanations.

7. L171 "Local stress measurements"

Did the authors distribute the brand-new gouge for the second experiment? Or did the authors just reset the location of the moving block and then reused the gouge layer that have sheared during the first experiment? Please clarify it.

8. L182-184 "These front propagation ... Cascadia subduction zone." L244-246 "Our observed front ... tectonic tremor sources."

I don't understand why the authors emphasize the consistency of the creep propagation speeds observed in the current experiments and in nature. The situations where the creep takes place are quite different between them (e.g. pressure, temperature, materials of gouge, water contents, scale, etc.). I cannot find any meaning other than a coincidence.

9. L195-202 "The sensitivity to ... of A1 events."

The terms "stronger", "weaker", and "strength" are too abstract. Does the "stronger event" mean the event with a large amount of slip? Please specific the physical meaning.

10. L25 "strength of creep fronts" L251-252 "creep front strength"

Please clarify the physical meaning of "strength."

11. L284-286 "Slip between the ... (A1 to A2, Fig. 1a)."

Please describe the exact locations of the slip sensors here.

12. Figure 4

Please label each panel with the lowercase letters (a-f).

Reviewer #6 (Remarks to the Author):

This study focuses on the effects of creep fronts and complexity on earthquake sequences. Laboratory experiments are performed on a hybrid sample configuration featuring rock gouge along the frictional interface and a compliant material in the bulk. One of the main findings of this study is that neighboring earthquakes are interconnected through creep fronts, which are highly sensitive to the stress levels left by previous earthquakes. These findings may help explain the spatial extent of the creep fronts and the relation between their propagation speed and the rheology of the hosting fault.

There were mainly two lines of reviewers' comments in the initial review of the manuscript: (i) Justification of the conclusions was scant due to the lack of local stress measurements; (ii) Model did not match the experimental setup. These issues have been addressed in the revision: a new experiment is included featuring measurements of local shear stress that are compared with creep front velocity measurements. Additional modelling shows the effect of the vertical dimension and the free edge. While these revisions have enhanced the manuscript, additional issues have emerged in the revised version of the manuscript. The new revision has resolved the remaining issues and I think is nearly ready for publication. The main points are discussed below:

- Creep sensitivity to local stress (Figure 4c-f). One of the issues raised by reviewer#2 about Figure 4c-f and the text associated to it was that the correlation between v_{cf} and v_{tr} and the initial overstress was not clear. This point has now been clarified by introducing trend lines and R^2 values. The new figure shows that there is correlation between initial overstress and v_{cf} and v_{tr} , particularly for the data that presents a range of event sizes and associated creep front velocities, but not between $A1$ stress drop and v_{cf} and v_{tr} . The related point on where the orange and red markers come from has also been addressed in the new Supplementary Figure 13, which explains the difference between the two different regimes "Two-period cycles" and "Variety of event sizes".

- Numerical model. There were several features of the model that needed to be addressed, namely the effect of the finite vertical dimension vs. infinite model space, the effect of the free surface and uniform shear stress loading. These aspects have been addressed by the authors and incorporated in the revised manuscript.

- Creep fronts associated with stress drops from $A1$ events. There is a statement in the manuscript reading: "Stress drops from $A1$ events drive creep fronts that gradually become more defined" but it was not clear what measurements supported this statement. Supplementary Figure 4 added in the revision helps to support this statement.

- There was also another point regarding placing this work in the context of existing theories. The authors have now introduced and/or extended discussion on models of fluid injection and induced seismicity (Battacharya and Viesca, *Science*, 2019; Wynants-Morel et al., *Journal of Geophysical Research*, 2020; Yang and Dunham, *Journal of Geophysical Research*, 2021; Garagash, *Philosophical Transactions of the Royal Society A*, 2021) and of afterslip associated to large earthquakes (Perfettini and Ampuero, *Journal of Geophysical Research*, 2008; Ariyoshi et al., *Tectonophysics*, 2019). In particular, Supplementary Figures 15 and 16 compare the measurements performed in this study to the models of Garagash, 2021 and Ariyoshi et al., 2019, respectively.

I have some additional comments:

- From an experimental point of view, the idea of using a hybrid sample, whose elasticity is controlled by compliant plastic forcing blocks but whose friction is dictated by a shear zone composed of a geological material (quartz gouge), is quite interesting and useful to investigate a problem that would be difficult to study with natural rock blocks. The idea is not completely new though as it has been already used in the studies by Buijze et al. *Journal of Geophysical Research* 2020 and Buijze et al., *Earth and Planetary Science Letters*, 2021, which were appropriately cited, and more recently by Rubino et al., *Nature* 2022, which should also be mentioned. While the experimental configuration studied in this work differentiates itself from those of Buijze et al. and Rubino et al., it is worth mentioning both those variants in this context.

- What do the authors believe is the effect of grain size evolution especially considering that they observe the creep behavior over large values of accumulated slip? Could it be that the evolution of friction properties and slip behavior is due, at least partly, to grain comminution?

- Repeatability of the experiments and other effects. "We conducted a suite of experiments varying σ_N , gouge layer mineralogy, and loading procedure and report here on one representative experiment on pure quartz gouge [...]" . Elaborating on the effect of normal stress, gouge layer mineralogy and loading procedure could enrich the current discussion. Can the authors summarize these effects? Otherwise, I am not sure it is worth mentioning that these effects have been explored if they are not discussed. Most importantly, what is the repeatability of the results presented in this study? Although only 1-2 representative experiments are described in detail, it would be appropriate to indicate that they are selected from a larger population of experiments (summarized in a table) leading to the same conclusions.

Reviewers Comments are in plain text. Author responses are in blue. Quoted text is shown in red.

Thanks to both reviewers for the careful reading and comments. In addition to many small changes throughout, we have added a more complete discussion of asperities and overstress to the main text, and we have added Supplementary Figs. 17-18 and Supplementary Table 2.

Reviewer #5 (Remarks to the Author):

Review of "Creep fronts and complexity in laboratory earthquake sequences illuminate delayed earthquake triggering" by Cebry et al. submitted to Nature Communications

This paper describes laboratory experiments that simulate a transition of fault slip behaviors with shear displacement and a delayed triggering on a gouge-filled laboratory fault. The experimental data and a numerical simulation suggest that velocity of creep front causing the delayed triggering is sensitive to shear stress level.

I enjoyed reading the manuscript very much. I think that this study is designed well and original. The experimental setting is noble, and the obtained data is valuable. However, I have some concerns as below, which should be addressed before publication.

[Major comments]

1. Asperity

1-1. Definition of asperity

As well known, definition of the "asperity" differs depending on the research theme (e.g. the area where really contacts, the area radiating the high-frequency seismic wave, the area slipped a lot during the earthquake, etc.). Therefore, unless it is clearly defined first, the readers including me cannot decide whether the authors' data interpretation and the model based on it is reasonable or not. It seems that the authors treat the high "normal" stress region as the asperity based on the FEM calculation (Supplementary Fig. 2), while the high "shear" stress is assumed for the asperity in the numerical simulation without any reasoning. Although the assumption should be basically reasonable as far as the friction coefficient is uniform over the fault, confusion and/or misunderstanding can be caused unless detailed explanation is added. To avoid such troubles, the authors should clearly define the physical meaning of the asperity, including the second asperity in L131 and others, in this study before starting the discussion.

We agree with the reviewer that there can be ambiguity surrounding the term asperity. We defined the term "seismic asperity" when it was first introduced on line 28-29 as "sections of the fault that are stronger and more unstable than their surroundings". However, we added additional disambiguation (from contact junction) when we discuss the asperities specific to our sample. That section now reads:

"The experiment exhibits two main seismic asperities at the forcing end (A1, $x = 0$) and leading end (A2, $x = 0.76$ m) of the sample (Fig. 1a). We describe the sample ends as seismic asperities because they are more unstable than their surroundings. The asperities have locally high σ_N due to a mechanical edge

effect common to biaxial sample configurations⁵²⁻⁵⁴ (Supplementary Fig. 2). This causes them to accumulate higher shear stress (Supplementary Figs. 17 and 18), and slip faster and have greater stress changes than the center part of the sample. The free surface at the end of the sample also reduces their stiffness and enhances instability. Note that the above definition is different from micron-scale junctions that compose a frictional interface (e.g. Dieterich and Kilgore, 1994) that are sometimes also referred to as asperities.”

1-2. Evidence for definition

Related to 1-1, direct evidence for the definition of the asperity should be provided. Although the authors may argue that the local normal stress cannot be measured, I believe that they can show the local shear stress distribution by using the strain gauge data, which is necessary for the readers to accept the authors' model. I also think that the strain data can help us understand what the second asperity is.

We have added Supplementary figures 17 and 18 to provide evidence to support our claim that the ends of the sample have both high normal stress and high shear stress and are more unstable than their surroundings. These are summarized here and copied below:

Supplementary Fig 17a shows that shear stress is somewhat higher on the fault ends than in the neighboring regions. However, these measurements also show a region of high shear stress near the center of the sample, and we believe that these absolute measurements of strain are not nearly as reliable as relative measures of strain. Supplementary Fig 17b shows relative measures of strain (strain rates) that indicate that shear stress is increasing at a faster rate near the sample ends than near the center of the sample during time periods when no significant slip events occur. These shear stress rates are quite stable over time. The faster rates at the sample ends indicate that more shear stress has accumulated there and that the fault is stronger (likely due to higher normal stress). Supplementary Fig 17c shows measured compaction of the quartz gouge layer at 7 locations along the length of the fault (from a different experiment, but still quartz gouge at 10 MPa normal stress), and those results show more shear-induced compaction near the ends of the samples, also suggesting higher normal stress there.

Supplementary Figure 18 shows some signs of wear (plastic deformation) on the teeth cut in the PMMA blocks near the sample ends. This wear has accumulated over many experiments (see list in new Supplementary Table 2), but shows that normal stress and shear stress reached higher levels near the sample ends than in the center of the sample, where no plastic deformation was observed.

Supplementary Fig. 17. Distribution of shear stress, shear stressing rate, and compaction along the sample. (a) Absolute shear stress measurements were made using a reference strain measurement when sample is not loaded. These measurements may not be entirely reliable because the strain and reference strain measurements were made many hours apart. (b) Shear stressing rate estimated over ≈ 10 s time intervals when the sample was being loaded and no slip events occurred. Different colors correspond to different time intervals to show stability over many stick-slip cycles. The faster rates at the sample ends indicate that more shear stress accumulates there and that the sample ends are stronger (they are able to carry more shear stress than the center of the sample). (c) Compaction measurements made by comparing gouge layer thickness after 5 mm of cumulative slip to that after 10 mm of cumulative slip at 10 MPa sample average normal stress. This shows that the ends of the sample compacted more than the center of the sample, consistent with the higher normal stress there.

Supplementary Fig. 18. Annotated photographs of the forcing end (left) and leading end (right) of the moving block taken after the experiment was completed, the two sample halves were separated, and the majority of the gouge was brushed off the fault (photo taken after experiment QS04_023, see Supplementary Table 2). The teeth machined into the PMMA block can be seen as vertical lines.

Compacted quartz gouge (white) is still stuck between the teeth, which is an indication that slip occurred within the gouge layer and not at the gouge/PMMA interface. There is evidence of plastic deformation of the PMMA teeth within 20-50 mm of the sample ends, while no deformation is observed closer to the center of the sample. The plastic deformation indicates that both shear and normal stress levels were higher near the sample ends (A1, and A2) than in the center of the sample.

2. Dependence of creep propagation speed on shear stress level

I understand that the main message from this study is that the creep propagation speed is sensitive to the shear stress level, which is individually suggested from the numerical simulation and the experimental data. But I have concerns about each of them.

2-1. Numerical simulation with simplified model setting

I have questions about the necessity of the numerical simulation in this study and the validity of the conclusions based on it. I agree that it is generally difficult and not always necessary to construct a perfect model which takes account of all the conditions to reproduce the real fault slip behaviors in the numerical simulation. However, it must be noted that we don't know how universal the calculated results are under the limited conditions. Regarding the current numerical simulation, the authors just investigate the creep propagation speed while varying only the shear stress level (at a X_{LP}). So, it's not surprising that those correlation was seen. For the universal conclusion, dependence of the creep propagation speed on other parameters should also be investigated. For example, although the asperities at both edges of the fault are modeled as the regions with locally high shear stress, the normal stress should also be high there as shown in Supplementary Fig. 2 and actually the high normal stress should cause the local high shear stress. According to the definition of L_b , higher normal stress makes L_b smaller, which should lead to more unstable.

Therefore, I wonder that the normal stress rather than the shear stress intrinsically characterizes the asperity and also controls the characteristics of the slip there (e.g. slip amount, slip rate, etc.). Since it should be natural to consider that the slip at the asperity affects the following creep behaviors, the normal stress may finally control the creep propagation speed.

We agree with the reviewer's comment that normal stress rather than shear stress intrinsically characterizes an asperity. It is also true that higher normal stress affects L_b , which was not taken into account with our simplified model with constant normal stress. However, those details primarily affect the asperity rupture, not the propagation of the creep fronts between the asperities. We focused on the propagation of creep fronts, irrespective of the asperity rupture, by keeping the asperity properties constant for all of our models. The reviewer is correct that our conclusions may not be entirely universal, as other factors, not varied in our models may be important. We focus on two parameters that did change in our experiment. We studied how creep front propagation speed is affected by (1) the initial shear stress level and (2) the friction parameters (See Fig. 3g). These are the parameters that primarily changed from one stick-slip event to another and over the course of our experiment, respectively.

The effects of normal stress on creep propagation speed were studied somewhat by Ariyoshi et al., 2018. However, the dependency on normal stress is more complicated since rate- and state-dependent friction effects are directly dependent on normal stress.

When we started this work, we also believed that the slip at the asperity (the “forcing”) primarily affects the creep behavior. This is why we kept the asperities constant for all of our simulations. However, the experimental results and comparison to theory (Garagash, 2021) suggest that under some circumstances the slip at the asperity does not play a primary role in the creep behavior. Instead, it is the on-fault stress state (and friction properties) in the region where the creep front propagates that primarily affects creep front characteristics further from the asperity. When the creep front propagates more than a few L_b from the asperity, the on-fault stress state dominates. Again, it is L_b of the region outside the asperity that matters not the L_b on the asperity. Our experiments generally agree with this interpretation.

It would be important to better understand, using a more realistic multi-cycle simulation, the environments when the “forcing” is more or less important compared to the initial stress levels. But that type of analysis requires a far more sophisticated modelling study that is outside the scope of the current work.

2-2. Overstress τ_0 in experiment

I don't understand why the deviation of the local shear stress from its long-term trend is defined as the overstress τ_0 in the experiment data. What is the physical meaning? Why isn't the absolute value used like the numerical simulation? I also wonder how the authors determined the gray trend lines shown in Figure 4b. Instead of the current one, how about using the deviation of the local shear stress at S6 and S7 from the temporal mean of the shear stress over the fault? I also suggest the authors to plot the shear stress data at S6 in Figure 4b because it is used as the parameter in Figure 4c.

We added the shear stress data at S6 in Figure 4b as suggested by the reviewer.

We didn't correlate creep front characteristics to the absolute value of stress because stress was slowly dropping over the course of the experiment (see Fig. 1c) as the fault weakened. We also could not correlate creep front characteristics to stress relative to the temporal mean of shear stress as the reviewer suggests because the temporal mean is a constant and the absolute stress is slowly dropping over the course of the experiment. Instead, we needed to choose a parameter that would represent the fault's stress state relative to its instantaneous steady-state strength, which is why we chose overstress, consistent with Garagash, (2021).

In our numerical models, on the other hand, absolute stress is proportional to overstress since the models started with a uniform initial sliding velocity and friction properties.

To clarify this difference between overstress and absolute stress, we defined separate symbols for overstress $\Delta\tau_0$ and absolute initial stress level τ_0 . We also added some text explaining the physical meaning for overstress and a discussion of how overstress and absolute stress are related in the model. That section now reads:

For the lab experiments, we compare creep front characteristics to the initial overstress $\Delta\tau_0$, not the absolute stress levels τ_0 , since different fault locations were observed to progressively strengthen or weaken with cumulative slip (Fig 4b). Overstress is the shear stress level relative to the fault's strength when sliding at a constant rate (steady state). Over the course of an earthquake cycle (or stick-slip cycle) the fault transitions from being above steady state (prior to the earthquake) to below steady state (just

after an earthquake), and this transition is facilitated by healing and breaking of frictional contacts (e.g. Fang et al., 2010). We define $\Delta\tau_0 \propto \tau_0 - \tau_{ss}$, where τ_{ss} is a reference level that changes linearly over time to match the long-term local strengthening or weakening trend over >10 stick-slip cycles (grey trend lines shown in Figure 4b). Note that in the numerical models described previously $\tau_0 \propto \Delta\tau_0$, since initial slip velocity V_{ini} and other parameters are uniform across the model.

3. Transition of slip behavior

One of the key features of this experimental study is observation of the transition of the slip behaviors with the shear displacement. The authors attribute it to the change in the frictional parameters with the evolution of shear fabric within the gouge layer. Although I can basically agree with the interpretation, I also have a concern. I think that a slip initiating at the asperity A1 must reach the forcing end of the fault. Once the slip reaches the end of the fault, the following slip behavior interacts with the apparatus system. Since the shear load is applied with the hydraulic jack, the oil volume within the jack increases with the shear displacement, which can greatly reduce the stiffness of the hydraulic jack and then leads to the decrease in the whole stiffness of apparatus. I just wonder that such decrease in the apparatus stiffness with the shear displacement can cause an apparent transition of the slip behavior.

It is true that A1 slip can be influenced by the stiffness of the loading machine. However, changes in the apparatus stiffness due to increased hydraulic fluid in the jack with increased cumulative displacement should not have an effect in this case. First, the maximum stroke of the jack is only about 13 mm. To achieve a cumulative slip of 25 mm (with the compliance of the sample) we had to retract the jack and add additional steel loading platens multiple times over the course of each experiment (see the “unload-reloads” noted in Figure 1c and Supplementary Figure 13). Second, we have measured the apparatus stiffness on this machine in previous studies (McLaskey and Yamashita, 2017; Cebry and McLaskey, 2021) and never found a notable difference in the stiffness as a function of cumulative slip, even when many additional steel loading platens were applied. The sample is always the most compliant element, so adding some additional steel or extending the cylinder has negligible effects on the overall stiffness.

4. Gouge layer

4-1. Uniformity of gouge distribution (L274-277 "The gouge layer ... 2.5 mm thickness.")

Given the significance of the effect of the gouge distribution on the slip behavior, it is necessary to describe how uniformly the gouge was distributed on the fault. Even with a small excess of the gouge, local normal and shear stress in the region can become higher than those in the surrounding region, which will form an asperity. I just suspect that the second asperity was generated by such a small heterogeneity of the gouge distribution.

We agree with the reviewer. Most likely secondary asperities were generated by imperfect distribution of gouge on the fault. To clarify this, we changed the description of the secondary asperities to read:

“The gouge layer and plastic blocks were prepared as uniformly as possible, but secondary asperities likely developed from small heterogeneity in the initial gouge distribution.”

4-2. Microstructure of gouge layer (L277-280 "Small teeth, ... plastic-gouge interface.")

Because the authors consider that the slip behaviors in the current experiment are governed by the frictional properties of the gouge, it is important to confirm that the slip did not take place at the interface between the PMMA block and the gouge but within the gouge layer. Did the authors investigate the microstructure of the sheared gouge layer after the experiment? Please show it if yes.

We did not perform a microstructural analysis of the gouge. Instead, we designed the teeth and gouge layer in an identical fashion to previous studies where both the friction properties and microstructure of the shear gouge layer were studied in detail (Scuderi et al., 2017), so we assume that both are similar in these experiments. Indeed, the sample behavior that we observe (transitions from slow slip to fast slip) confirms that the friction behavior and its evolution are quite similar to previous studies (Leeman et al., 2016; Scuderi et al., 2017), so we expect microstructure is also similar.

We did analyze the gouge after the experiments. The gouge had compacted into subparallel flakes, which were quite fragile. However, we found it very difficult to remove the gouge from between the PMMA teeth. That removal process required persistent brushing or scrubbing. These observations suggested that comminution and grainsize reduction had occurred on at least one shear zone within the gouge layer and not at the gouge/PMMA interface. We added Supplementary Figure 18 (copied in response to a comment above). It shows the quartz gouge compacted into the PMMA teeth. Hopefully it is quite clear that slip cannot occur along interface between the PMMA and gouge. The teeth force the slip to occur within the gouge layer.

[Minor comments]

5. L106-107 "The entire sample ... (Fig. 1c, 1d; Fig. 2a)."

Please clarify the meaning of "bifurcation" here. The "bifurcation" for the asperity A1 in L106 and that for A2 in L117 are the same thing? I think that the second one means the alternating slip styles between slow and fast on an asperity, which should be consistent with the slip styles called "bifurcation" in other studies (e.g. Veedu et al., 2020, GRL). But I don't understand what the authors exactly mean by that term in L106.

We agree with the reviewer that it is incorrect to describe the development of slow slip as a bifurcation, or at least it is inconsistent with previous descriptions of bifurcation. We changed this section to omit the word bifurcation and now only refer to the oscillations at A2 as a bifurcation.

6. L141-142 "the structures that ... that reinforce them."

What are "rheological changes"? Does it mean the changes in the frictional parameters? Please add some explanations.

Yes, we mean changes in friction parameters. We changed "rheological changes" to "changes to friction properties" and throughout the manuscript we have changed "rheology" with "friction".

7. L171 "Local stress measurements"

Did the authors distribute the brand-new gouge for the second experiment? Or did the authors just reset the location of the moving block and then reused the gouge layer that have sheared during the first experiment? Please clarify it.

We distribute brand new gouge for every experiment, and we added a note of this in the methods section.

8. L182-184 "These front propagation ... Cascadia subduction zone." L244-246 "Our observed front ... tectonic tremor sources."

I don't understand why the authors emphasize the consistency of the creep propagation speeds observed in the current experiments and in nature. The situations where the creep takes place are quite different between them (e.g. pressure, temperature, materials of gouge, water contents, scale, etc.). I cannot find any meaning other than a coincidence.

Certainly the conditions are different. However, we do not believe that the similarity in propagation speeds is mere coincidence. The shear wave velocity of the PMMA is similar (to an order-of-magnitude) to what is expected at depth in the earth, despite differences in temperature, pressure, material, and water content. This similarity is likely the controlling factor. If the propagation speed was much larger (in either lab or earth) compared to the shear wave velocity, it would not be slow creep, it would be dynamic rupture. If it were much smaller, it would not be detectable with standard tools and human time scales. So it may be a detection issue as well.

9. L195-202 "The sensitivity to ... of A1 events."

The terms "stronger", "weaker", and "strength" are too abstract. Does the "stronger event" mean the event with a large amount of slip? Please specific the physical meaning.

We added two sentences to clarify the meaning of "strong", copied below.

"In the above discussion, we refer to "strong" events as those that slip faster, have larger total slip amount, and have larger local stress changes. These parameters are directly correlated for events with rupture dimensions less than $\approx 5h^*$ (Wu and McLaskey, 2019). "

10. L25 "strength of creep fronts" L251-252 "creep front strength"

Please clarify the physical meaning of "strength."

By strength we mean the maximum slip velocity at the creep front. We added this clarification on this line. It now reads: "We find that outside a distance $\approx 5L_b$ from the location of hypocentral forcing, creep front strength (maximum slip velocity) and propagation velocity are extremely sensitive to fault stress levels..."

11. L284-286 "Slip between the ... (A1 to A2, Fig. 1a)."

Please describe the exact locations of the slip sensors here.

The sentence was changed to read: "...8 eddy current displacement sensors located along the length of the fault (at locations 30, 130, 230, 330, 430, 530, 630, and 730 mm from the forcing end (Fig. 1a). "

12. Figure 4

Please label each panel with the lowercase letters (a-f).

We have added labels to each panel.

Reviewer #6 (Remarks to the Author):

This study focuses on the effects of creep fronts and complexity on earthquake sequences. Laboratory experiments are performed on a hybrid sample configuration featuring rock gouge along the frictional interface and a compliant material in the bulk. One of the main findings of this study is that neighboring earthquakes are interconnected through creep fronts, which are highly sensitive to the stress levels left by previous earthquakes. These findings may help explain the spatial extent of the creep fronts and the relation between their propagation speed and the rheology of the hosting fault.

There were mainly two lines of reviewers' comments in the initial review of the manuscript: (i) Justification of the conclusions was scant due to the lack of local stress measurements; (ii) Model did not match the experimental setup. These issues have been addressed in the revision: a new experiment is included featuring measurements of local shear stress that are compared with creep front velocity measurements. Additional modelling shows the effect of the vertical dimension and the free edge. While these revisions have enhanced the manuscript, additional issues have emerged in the revised version of the manuscript. The new revision has resolved the remaining issues and I think is nearly ready for publication. The main points are discussed below:

- Creep sensitivity to local stress (Figure 4c-f). One of the issues raised by reviewer#2 about Figure 4c-f and the text associated to it was that the correlation between v_{cf} and v_{tr} and the initial overstress was not clear. This point has now been clarified by introducing trend lines and R^2 values. The new figure shows that there is correlation between initial overstress and v_{cf} and v_{tr} , particularly for the data that presents a range of event sizes and associated creep front velocities, but not between A1 stress drop and v_{cf} and v_{tr} . The related point on where the orange and red markers come from has also been addressed in the new Supplementary Figure 13, which explains the difference between the two different regimes "Two-period cycles" and "Variety of event sizes".

Correct. Thanks for the careful reading and accurate summary.

- Numerical model. There were several features of the model that needed to be addressed, namely the effect of the finite vertical dimension vs. infinite model space, the effect of the free surface and uniform shear stress loading. These aspects have been addressed by the authors and incorporated in the revised manuscript.

Correct.

- Creep fronts associated with stress drops from A1 events. There is a statement in the manuscript reading: "Stress drops from A1 events drive creep fronts that gradually become more defined" but it was not clear what measurements supported this statement.

Supplementary Figure 4 added in the revision helps to support this statement.

We are glad this figure helped.

- There was also another point regarding placing this work in the context of existing theories. The authors have now introduced and/or extended discussion on models of fluid injection and induced seismicity (Battacharya and Viesca, *Science*, 2019; Wynants-Morel et al., *Journal of Geophysical Research*, 2020; Yang and Dunham, *Journal of Geophysical Research*, 2021; Garagash, *Philosophical Transactions of the Royal Society A*, 2021) and of afterslip associated to large earthquakes (Perfettini and Ampuero, *Journal of Geophysical Research*, 2008; Ariyoshi et al., *Tectonophysics*, 2019). In particular, Supplementary Figures 15 and 16 compare the measurements performed in this study to the models of Garagash, 2021 and Ariyoshi et al., 2019, respectively.

I have some additional comments:

- From an experimental point of view, the idea of using a hybrid sample, whose elasticity is controlled by compliant plastic forcing blocks but whose friction is dictated by a shear zone composed of a geological material (quartz gouge), is quite interesting and useful to investigate a problem that would be difficult to study with natural rock blocks. The idea is not completely new though as it has been already used in the studies by Buijze et al. *Journal of Geophysical Research* 2020 and Buijze et al., *Earth and Planetary Science Letters*, 2021, which were appropriately cited, and more recently by Rubino et al., *Nature* 2022, which should also be mentioned. While the experimental configuration studied in this work differentiates itself from those of Buijze et al. and Rubino et al., it is worth mentioning both those variants in this context. We added reference to Rubino et al. (2022)

- What do the authors believe is the effect of grain size evolution especially considering that they observe the creep behavior over large values of accumulated slip? Could it be that the evolution of friction properties and slip behavior is due, at least partly, to grain comminution? Yes, absolutely. The evolution of friction properties is due, at least partly, to grain comminution, but the main effect is associated with shear localization. The work of Marone and Scholz (1989) documents carefully the changes in grain size distribution. They show that granular material with a uniform particle size distribution (monodisperse or well-sorted material) evolves toward a power-law size distribution over engineering shear strains of 3-5. Another relevant study is that by Marone and Kilgore (1993) who showed that comminution cannot explain the changes in RSF parameters with slip, but rather these changes are related to shear localization. Scuderi et al., (2017) presented evidence for this on identical gouge layers deformed in a different apparatus. To emphasize this point, we have changed this section to read:

“With accumulated shear strain, granular gouge and fault rock evolves from velocity strengthening to velocity weakening friction and \$D_c\$ decreases (Marone and Kilgore, 1993). These changes are due primarily to shear localization rather than reduction of particle size. ... Scuderi et al. (2017) described this as an evolution from distributed deformation throughout the gouge layer to localized deformation along fault parallel shear planes, and showed evidence for

comminution and grain size reduction. We expect similar behavior in our experiments since they utilize identical gouge layers (see Methods).”

- Repeatability of the experiments and other effects. “We conducted a suite of experiments varying σ_N , gouge layer mineralogy, and loading procedure and report here on one representative experiment on pure quartz gouge [...]”. Elaborating on the effect of normal stress, gouge layer mineralogy and loading procedure could enrich the current discussion. Can the authors summarize these effects? Otherwise, I am not sure it is worth mentioning that these effects have been explored if they are not discussed. Most importantly, what is the repeatability of the results presented in this study? Although only 1-2 representative experiments are described in detail, it would be appropriate to indicate that they are selected from a larger population of experiments (summarized in a table) leading to the same conclusions.

We added Supplementary Table 2, which shows the chronology of all of the experiments made with this experimental system. We focused our more recent work only on quartz gouge layers, since those produced behavior that was both interesting and repeatable. We added the following sentences to the methods to very briefly indicate the repeatability of the study and the behavior of the different mineralogy.

All six experiments on quartz showed a similar evolution of behavior that culminated with partial ruptures and variable delayed triggering, though not all experiments were loaded at a smooth and constant rate or instrumented as completely. Talc produced only slow slip, and the behavior of gypsum was less repeatable.

REVIEWERS' COMMENTS

Reviewer #5 (Remarks to the Author):

Review of revised version of "Creep fronts and complexity in laboratory earthquake sequences illuminate delayed earthquake triggering" by Cebry et al. submitted to Nature Communications

I have raised many issues for the previous version of this paper. Now I found that most of them are fully addressed. Although the replies to rest of them are based on indirect evidence and thus imperfect, I also understand the difficulties of the laboratory experiment and I am satisfied with the authors' efforts. In total, I believe that now this paper is ready for publication.

REVIEWERS' COMMENTS

Reviewer #5 (Remarks to the Author):

Review of revised version of "Creep fronts and complexity in laboratory earthquake sequences illuminate delayed earthquake triggering" by Cebry et al. submitted to Nature Communications

I have raised many issues for the previous version of this paper. Now I found that most of them are fully addressed. Although the replies to rest of them are based on indirect evidence and thus imperfect, I also understand the difficulties of the laboratory experiment and I am satisfied with the authors' efforts. In total, I believe that now this paper is ready for publication.

Thank you for reading the paper again. We are glad that you are satisfied with the responses.

To clarify the indirect nature of some of the evidence, we added a sentence to the caption of Supplementary Figure 17 that reads: "Note that the compaction measurements were made in a different experiment (QS04-020) with similar conditions, described in Supplementary Table 2."

We also added the following sentence to the caption of Supplementary Figure 18, describing the caveat: "Note that the deformation reported here is the result of the cumulative effect of all experiments reported in Supplementary Table 2."